# Podocyte Death in Diabetic Kidney Disease: Potential Molecular Mechanisms and Therapeutic Targets

**DOI:** 10.3390/ijms25169035

**Published:** 2024-08-20

**Authors:** Suye Zhong, Na Wang, Chun Zhang

**Affiliations:** Department of Nephrology, Union Hospital, Tongji Medical College, Huazhong University of Science and Technology, Wuhan 430022, China; m202376137@hust.edu.cn (S.Z.); wnflora_2011@163.com (N.W.)

**Keywords:** podocyte, cell death, apoptosis, autophagy, necroptosis, pyroptosis, ferroptosis, diabetic kidney disease

## Abstract

Cell deaths maintain the normal function of tissues and organs. In pathological conditions, the abnormal activation or disruption of cell death often leads to pathophysiological effects. Diabetic kidney disease (DKD), a significant microvascular complication of diabetes, is linked to high mortality and morbidity rates, imposing a substantial burden on global healthcare systems and economies. Loss and detachment of podocytes are key pathological changes in the progression of DKD. This review explores the potential mechanisms of apoptosis, necrosis, autophagy, pyroptosis, ferroptosis, cuproptosis, and podoptosis in podocytes, focusing on how different cell death modes contribute to the progression of DKD. It recognizes the limitations of current research and presents the latest basic and clinical research studies targeting podocyte death pathways in DKD. Lastly, it focuses on the future of targeting podocyte cell death to treat DKD, with the intention of inspiring further research and the development of therapeutic strategies.

## 1. Introduction

Diabetic kidney disease (DKD) is a leading contributor to rising mortality trends [1], accounting for approximately 50% of cases of end-stage renal disease (ESRD) in developed countries [2,3]. Patients with ESRD typically require long-term dialysis or kidney transplantation, which is associated with high medical costs [4,5,6]. In clinical settings, the diagnosis of DKD is based on the presence of albuminuria exceeding 300 mg/d and/or a reduced estimated glomerular filtration rate (eGFR) below 60 mL/min/1.73 m^2^ in diabetic patients, while concurrently excluding any alternative chronic kidney disease (CKD) that might mimic its clinical manifestations and paraclinical findings [7,8]. By 2050, the global diabetes burden is estimated to reach 1.31 billion cases, and the age-standardized total diabetes prevalence rate is projected to exceed 10% [9]. The International Diabetes Federation (IDF) has projected that approximately 40% of individuals with diabetes are expected to progress towards developing DKD [10]. DKD was responsible for 13.09 million disability-adjusted life years (DALYs) in 2019 [3], revealing a widespread and concerning trend on patients’ quality of life by increasing the risk of kidney failure, accelerating the progression of cardiovascular (CV) diseases, which include heart failure (HF), myocardial infarction (MI), stroke, and CV death [11], and contributing to premature mortality [12,13].

Among the various glomerular morphological features, microscopic structural features reveal enlarged podocytes, increased subpodocyte space, and reduced podocyte numbers within each glomerulus serving as indicative signs of progressive DKD [14]. As DKD progresses, the podocyte count decreases, thereby making podocyte loss as one of the earliest cellular hallmarks in human DKD. Any disruption of podocyte function, including hypertrophy, desmosomal detachment, decreased cell density, and apoptosis, can undermine the barrier’s integrity, resulting in heightened proteinuria, impaired renal function, and elevated creatinine concentrations [15]. Current therapeutic strategies for podocytes rely on repurposed immunosuppressive drugs [16]. In light of this, the exploration and development of innovative approaches aimed at regulating cell death pathways to maintain podocyte numbers, thereby protecting against or managing DKD, holds a promising future.

In recent years, research focused on elucidating the impact of podocyte injury in DKD has intensified. Studies have uncovered that a multitude of factors, such as renal blood flow dynamics, metabolic disorders, inflammation, cellular senescence, epigenetic regulations, epithelial-mesenchymal transition (EMT), and cell death pathways, may contribute to podocyte injury [17,18,19,20,21,22,23,24,25], sparking a surge in efforts to develop targeted therapeutic strategies specifically for this condition. The expression of cell death-related genes, including Bid, Dapk1, and Cd40, was significantly elevated in diabetic podocytes. There was also a modest increase in cleaved caspase-3, indicating augmented podocyte apoptosis in type 1 diabetic nephropathy (T1DM) [26]. Additionally, the downregulation of genes primarily associated with RNA processing and endoplasmic reticulum function in diabetic podocytes suggests potential modifications in the mTOR pathway and autophagy/endoplasmic reticulum stress response, both of which are implicated in podocyte injury in type 2 diabetic nephropathy (T2DM) [27]. However, the exact mechanisms underlying podocyte damage of DKD remain to be fully elucidated.

The high glucose (HG) induced by diabetes stimulates the activation of the renin-angiotensin-aldosterone system (RAAS) [28], upregulates sodium-glucose cotransporter-2 (SGLT-2) receptor [29,30,31], and secretes endothelin-1 (ET-1) [32]. This alters renal blood flow dynamics, ultimately leading to a sustained rise in glomerular filtration rate (GFR) and the onset of proteinuria [33].

Hyperglycemia not only induces the polyol pathway, protein kinase C (PKC), and the hexosamine pathway, but it also leads to the formation of advanced glycation end products (AGEs), oxidative damage, and hypoxia. These factors result in kidney hypertrophy and the progression of DKD [34]. AGEs are recognized as a biomarker of biological aging and may contribute to renal lesions in DM relate to kidney aging [35]. Moreover, AGEs promote to the expression of transforming growth factor-β (TGF-β) in podocytes, playing a significant role in the development of glomerulosclerosis and tubulointerstitial fibrosis in DKD [36].

Endothelial dysfunction is a prominent feature in DKD [37]. Renal glomerular endothelial cells, characterized by numerous fenestrations and rich glycocalyx, play a crucial role in maintaining structural changes in podocytes and the glomerular filtration barrier (GFB) [38]. Endothelin-1 (ET-1) can directly activate the nuclear factor kappa-B (NF-κB) and β-catenin signaling pathways in podocytes, thereby promoting the progression of diabetic glomerulosclerosis and leading to podocyte loss [39]. In addition, the activation of the ET-1 receptor A is intricately linked to podocyte damage [32]. The endothelial glycocalyx (eGlx) is a polymeric, sugar-rich network that lines the surface of endothelial cells, serving to protect them [40]. Heparan sulfate (HS) is a prominent component of the eGlx; however, heparinase, an enzyme responsible for cleaving HS, is highly expressed in the kidney in DKD. The lack of HS plays a causative role in reducing endothelial glycocalyx depth. It has been observed that eGlx-dependent glomerular albumin permeability is elevated in diabetes [41]. Podocytes also play a significant role in enhancing the synthesis of heparanase, which in turn contributes to glycocalyx loss [42]. DKD originates from the detrimental effects of intracellular hyperglycemia, impairing the endothelium through a complex interplay of inflammatory responses, disrupted communication between endothelial cells and podocytes, as well as the involvement of exosomes [43,44,45].

DKD is also a heterogeneous and polygenic disease. Genetic susceptibility and environmental exposure play a pivotal role in its progression [46]. Human podocytes show changes in DNA methylation when exposed to HG [47]. For instance, epigenetic regulators such as HDACs (histone deacetylases) and DNMTs (DNA methyltransferases) modulate the inflammatory response and apoptosis of podocytes exposed to a high-glucose environment, ultimately contributing to early proteinuria in DKD [48]. Moreover, diabetic patients frequently exhibit a phenomenon known as metabolic memory, where the pathological changes triggered by previous hyperglycemia persist even after the subsequent stabilization of glucose control [49,50]. Histone lysine methylation participates in the regulation of key genes associated with fibrosis and the inflammation of DKD, persisting in vascular cells irrespective of a high-glucose environment [51]. This may partly explain the “metabolic memory” phenomenon observed in response to HG [52].

In an HG environment, there is an increase in the production of reactive oxygen species (ROS) and heightened nicotinamide adenine dinucleotide phosphate (NADPH) oxidase activity in podocytes has been observed. Concurrently, the levels of essential antioxidants, such as glutathione (GSH) and superoxide dismutase (SOD), show a decline, which compromises the integrity of the podocyte’s antioxidant defense system [53]. Inflammation arises from the interaction between inflammatory mediators released by local and circulating immune cells and renal cell populations [54]. In an oxidative environment in DKD, the activation of NF-κB leads to the release of substantial amounts of proinflammatory cytokines and the recruitment of inflammatory cells [55]. It is evident that the interplay between oxidative stress and inflammation in the pathophysiology of DKD creates a vicious cycle of oxidative stress-induced inflammation, causing damage to structural integrity and function of the kidneys [56]. Additionally, mitochondrial dysfunction contributes to increased oxidative stress and activation of inflammatory pathways, ultimately resulting in progressive renal function decline and fibrosis [57].

A high-glucose environment promotes premature cellular senescence in various cell types [58]. In DKD, metabolic disturbances lead to excessive production of ROS by NADPH oxidase in podocytes, culminating in cellular senescence [59]. Notably, cellular senescence may serve as an early diagnostic indicator for DKD [60]. Additionally, FSP1 (fibroblast-specific protein 1) is constitutively expressed in the cytoplasm of fibroblasts or epithelial cells undergoing EMT in tissues. The presence of FSP1 in podocytes among diabetic patients is associated with aggravated clinical and pathological manifestations of DKD, potentially due to EMT-induced detachment of podocytes [61]. The diverse modes of podocyte death include apoptosis, autophagy, pyroptosis, necroptosis, ferroptosis, and cuproptosis (Figure 1). These pathways are interconnected and can be activated concurrently or independently [62,63]. Podocyte injury is also crucial in the progression of DKD [14]. It should be noted that the specific cellular and molecular pathophysiological mechanisms triggering podocyte death, remain elusive. This review provides comprehensive insights into the various patterns of cell death, underlying the mechanisms and therapeutic strategies in DKD. This will deepen our understanding of podocyte demise and lay the groundwork for the development of novel and ideal treatment approaches for DKD.

## 2. The Podocyte Deaths That Trigger the Progression of DKD

### 2.1. Autophagy

Autophagy serves as the fundamental intracellular degradation process, directing the cytoplasmic materials towards lysosomal degradation to preserve cellular homeostasis and renew cellular components [64], as depicted in Figure 2. It exhibits two distinct modes: non-selective, involving in the degradation of a wide range of cellular components, and selective, specifically targeting organelles, ribosomes, and protein aggregates for degradation [65,66,67]. Current research has identified three forms of autophagy: chaperone-mediated autophagy, microautophagy, and macroautophagy, each playing important roles in cellular homeostasis and survival [68]. Autophagy-dependent cell death can be defined as a form of regulated cell death (RCD) that depends on the autophagic machinery [65], which is mediated by evolutionarily conserved autophagy-related genes (*Atg*) [69,70]. Under diabetic conditions, a decline in *Atg3* expression within podocytes is observed. This disrupts the autophagy flux and facilitates the transition of the slit diaphragm (SD) structure to a tight junction (TJ) structure, causing foot process effacement [71].

Basal autophagy is a continuous process in virtually all eukaryotic cells, essential for maintaining cellular homeostasis by eliminating surplus or damaged organelles and long-lived proteins. However, whenever intracellular or environmental equilibrium is disrupted, there is a significant increase in the intensity of the autophagic response, leading to its augmentation [72]. This heightened autophagic activity can directly implicate cell death and contribute to the pathogenesis of various human diseases [68], including kidney disorders [73]. Many researches indicate that autophagy plays a crucial role in preserving lysosomal homeostasis in podocytes under diabetic conditions, and its dysfunction is instrumental in the pathogenesis of podocyte loss, ultimately resulting in increased proteinuria in DKD [74]. Suppressed podocyte autophagy is one of the key features in DKD [75,76,77].

The primary signaling pathways that promote the progression of DKD involve autophagy in podocytes, including the mammalian target of rapamycin (mTOR), AMP-activated protein kinase (AMPK), phosphatidylinositol-3-kinase (PI3K)/protein kinase B (Akt), extracellular regulated kinase/mitogen-activated protein kinase (ERK/MAPK), c-Jun N-terminal kinase (JNK), and Janus protein tyrosine kinase/signal transducer and activator of transcription (JAK/STAT) pathways [78,79,80,81,82,83]. In addition, the Jiedu Tongluo Baoshen formula effectively enhances podocyte autophagy and reduces proteinuria in DKD by inhibiting the PI3K/Akt/mTOR signaling pathway [84]. HG levels significantly augment the autophagic flux in the glomeruli of diabetic mice and in both primary and immortalized podocyte cultures [85]. In streptozotocin-induced diabetic mice, a decline in basal autophagy is evident, as evidenced by the decreased levels of beclin 1, ATG12-ATG5 complex, and LC3-II, providing crucial insights into the autophagic process in the context of diabetes [86]. Insulin has been observed to counteract the decline in autophagy-related proteins, including ATG5, Beclin1, and LC3B, as well as podocyte-specific markers such as podocin, nephrin, and synaptopodin in diabetic kidney tissues [87]. Risa, a regulator of insulin sensitivity and autophagy, was found to improve the autophagy levels in renal podocytes through inhibition via the Sirt1/GSK3β (glycogen synthase kinase-3β) pathway, thereby promoting the progression of DKD [88]. Based on these observations, it is reasonable to speculate that hyperglycemia contributes to a reduction in podocyte autophagy, thereby accelerating the development of DKD. The potential of insulin to alleviate this process is attributed to its capacity to enhance autophagy.

Notably, the sirtuin family is involved in the regulation of autophagy in podocytes. Specific knockdown of Sirt6 in podocytes exacerbates podocyte damage and proteinuria in DKD [89]. Sestrin2 upregulation was found to promote autophagy and inhibit apoptosis in DKD. This observation was substantiated by the ameliorative effects of Sestrin2 overexpression, which led to improved podocyte integrity, mesangial proliferation, proteinuria, and RF in experimental animal models of DKD [90]. The deletion of Sirt6 in podocytes specifically exacerbated podocyte damage and proteinuria in DKD mouse models, with its mechanism involving the downregulation of Notch1 and Notch4 gene transcription by deacetylating H3K9, thereby protecting podocytes from apoptosis and inflammation by increasing autophagic flux [91]. Additionally, sirtuin activators exhibit protective properties in the context of DKD [92].

UPS and autophagy are the two primary pathways for protein degradation within cells. The UPS selectively targets proteins with specific polyubiquitin modifications for degradation by the proteasome, while autophagy plays a crucial role in mediating the degradation of ubiquitinated protein aggregates [93]. Furthermore, there is crosstalk between UPS and autophagy-mediated by the unfolded protein response (UPR). Inhibition of the proteasome can directly impact autophagy-related proteins, leading to a compensatory upregulation of autophagy. Conversely, the inhibition of autophagy can either activate or impair proteasomal flux through various mechanisms [94]. Recent studies have consistently highlighted the pivotal roles of protein ubiquitination and deubiquitination in the regulation of autophagy [95]. During selective autophagy, lipidated ATG8 proteins participate in cargo selection by binding to autophagy cargo receptors, which can recognize ubiquitinated cargo. Gradually, the membrane structure expands to form a cellular organelle called an autophagosome, eventually fusing with a lysosome [96]. The autophagy-lysosome pathway (ALP) serves as a crucial compensatory mechanism for facilitating the breakdown of ubiquitinated protein aggregates. The lysosomal-dependent autophagic pathway plays a pivotal role in podocyte homeostasis in DKD [97]. Additionally, mitochondria have been described to be involved in the sensing of UPS impairment and in AMPK-mediated upregulation of autophagy. In podocytes, autophagy is mainly regulated through the AMPK-ULK1 axis rather than inhibition of MTOR [85]. Previous research has demonstrated that the deubiquitinating enzyme USP11 promotes autophagy by activating the AMPK/Akt/mTOR signaling pathway, leading to ULK1 activation and the initiation of autophagy, thereby playing a role in regulating autophagy in colorectal cancer. [98] Given the involvement of USP11 and ULK1 in autophagy regulation, it can be inferred whether these two molecules interact and participate in podocytes, and the progression of DKD. Thus, an interplay between the UPS and the ALP in podocytes is promising for the development of DKD, but current empirical evidence remains inconclusive, necessitating further investigation in future studies [95].

### 2.2. Pyroptosis

Pyroptosis is a form of inflammatory programmed cell death (PCD), characterized by the activation of gasdermin D (GSDMD) and caspases (caspase-1/4/5/11) [99,100]. The activation of proapoptotic caspase-3 also triggers pyroptosis by cleaving GSDME [101]. Furthermore, granzyme has been found to induce caspase-independent pyroptosis in cells expressing either GSDME or GSDMB. The morphological characteristics of cell pyroptosis include cell swelling, membrane blebbing, DNA fragmentation, and eventual cell lysis. However, the nucleus typically remains intact, distinguishing it from the nuclear damage observed in apoptosis and necrotic cell death [102]. It is widely accepted that pyroptosis can cause sterile inflammation activated by extracellular or intracellular stimulation such as bacteria, viruses, toxins, and chemotherapy drugs [103]. Pyroptosis is primarily orchestrated by pattern recognition receptors (PRRs) within the innate immune system, such as nod-like receptors (NLRs) and AIM2-like receptors (ALRs). These receptors detect and respond to danger-associated molecular patterns (DAMPs), pathogen-associated molecular patterns (PAMPs), or disruptions in cellular homeostasis, leading to the activation of caspases. Upon activation, these caspases cleave the GSDMD protein, initiating the process of pyroptotic cell death [104]. The activation of diabetes-related mediators, including toll-like receptor 4 (TLR4), NLRP3, caspase-1, IL-1β, IL-18, and GSDMD-NT [105], plays a pivotal role in the pathophysiology of DKD (Figure 3).

Under HG cultivation, human and mouse podocytes exhibited the activation of caspase-11/4- and GSDMD-dependent pyroptosis. Suppression of this pyroptosis may prevent podocyte loss and local inflammation, thereby attenuating the progression of DKD [106], as shown in Figure 3. This suggests a significant upregulation of pyroptosis in the podocytes of DKD, which might implicate targeting key molecules of the necroptotic pathway or inflammatory mediators as a potential strategy to slow disease progression.

Some chemical compounds facilitate podocyte pyroptosis by promoting the expression of NLRP3. A recent study has shown that pyroptosis in podocytes can be mitigated by modulating the miR-155-5p/HO-1/NLRP3 axis, against which Dapagliflozin applies its protective effect [107]. Research suggests that rutaecarpine exhibits a protective role in DKD by specifically targeting the VEGFR2/NLRP3 signaling pathway, thereby preventing or mitigating the inflammatory and pyroptosis that contribute to podocyte damage [108]. Moreover, a recent investigation reveals that lysophosphatidic acid (LPA) triggers NLRP3 inflammasome activation through downregulating the enhancer of Ezh2 and H3K27me3 histone modifications, alongside upregulation of Egr1 expression. This is ultimately conducive to podocyte injury and pyroptosis, which may be a potential mechanism of DKD progression [109]. Additionally, flavones extracted from a plant known as Abelmoschus Manihot (TFA) could ameliorate HG-induced pyroptosis and injury in podocytes in DKD by modulating methyltransferase-like 3 (METTL3)-regulated m6A modification. This modulation influences NLRP3-inflammasome activation and regulates the phosphatase and tensin homolog (PTEN)/PI3K/Akt signaling pathway [110]. In addition, the stimulator of interferon genes (STING), driving noninfectious inflammation and pyroptosis, may be a potential target for podocyte injury in DKD [111]. Administration of N-acetylmannosamine (ManNAc) was found to protect podocytes from pyroptosis in mice with DKD by mitigating mitochondrial dysfunction and suppressing the ROS/NLRP3 signaling pathway [112]. Fucoidan has been substantiated in research to target NLRP3 inflammasome-induced podocyte pyroptosis, thereby attenuating the development of RF in DKD patients [111]. Finally, catapol may effectively suppress oxidative stress and inflammation associated with pyroptosis. Its underlying mechanism could be associated with the AMPK/SIRT1/NF-κB signaling pathway. These findings suggest that catapol holds promising therapeutic potential for DKD [113]. The collective findings indicate a significant role of podocyte pyroptosis in the pathogenesis of DKD.

### 2.3. Apoptosis

Apoptosis, an active and genetically programmed process in which caspase activation plays a central role [114], is known to be modulated by a diverse array of environmental factors encompassing both physiological and pathological stimuli [115]. The morphological characteristics of apoptosis involve cell shrinkage and pyknosis, with intact cellular organelles, subsequently followed by the formation of membrane-bound protrusions that can be separated. Within tissues, these structures are subsequently phagocytosed by macrophages, parenchymal cells, or neoplastic cells and ultimately degraded within phagolysosomes, without any associated inflammation [114,116].

The primary regulatory pathways of apoptosis can be summarized as the extrinsic pathway or death receptor pathway and the intrinsic pathway or mitochondrial pathway, which interact with molecules that mutually influence each other [117] (Figure 4).

Numerous studies have been dedicated to investigating the mechanism of podocyte apoptosis in DKD, highlighting it as the predominant mode of podocyte death [62]. Khazim et al. demonstrated that exposure of podocytes to an HG concentration resulted in a 150% increase in apoptotic podocyte death [90]. In an HG environment, the transcription factor p53 plays a crucial role in regulating podocyte apoptosis, autophagy, and ferroptosis mechanisms, thereby mediating podocyte injury and promoting the progression of DKD [132,133]. For instance, the upregulation of retinoic acid receptor responder protein-1 (RARES1) has been observed in podocytes in a DKD mouse model. Matrix metalloproteinase 23 (MMP23), which is highly specific to podocytes, cleaves RARES1 to produce its soluble form, sRARES1. This soluble form subsequently enters podocytes via endocytosis and contributes to enhanced podocyte apoptosis, facilitated by p53, ultimately leading to aggravated glomerular injury and renal dysfunction. Therefore, targeting the specific functions of RARES1 and MMP23 in podocytes may hold promise as a therapeutic strategy for treating DKD [134]. Recent research has emphasized the therapeutic potential of modified Hu-lu-ba-wan (MHLBW) in the management of DKD, as it effectively suppresses glomerular injury and prevents podocyte apoptosis by positively regulating Pyruvate kinase isozyme type M2(PKM2)-mediated mitochondrial homeostasis [135]. In a recent experiment, Carnitine palmitoyltransferase-1A (CPT1A), a key rate-limiting enzyme in fatty acid oxidation (FAO), has been shown to protect podocytes from lipotoxicity and apoptosis in vitro, and demonstrates promising therapeutic benefits in vivo for DKD [136]. Furthermore, recent findings have elucidated the molecular mechanism through which insulin-like growth factor-binding protein 2 (IGFBP2) contributes to podocyte apoptosis in the progression of DKD, involving the induction of mitochondrial damage via the ITGA5/FAK phosphorylation pathway [137]. Additionally, the most recent study has demonstrated that miR-4645-3p exhibits a protective impact on podocyte injury and mitochondrial dysfunction in DKD by specifically targeting cyclin-dependent kinase 5 (Cdk5) [138]. As previously mentioned, addressing key elements of abnormal glycolysis, dyslipidemia, and mitochondrial dysfunction has the potential to alleviate podocyte apoptosis, thereby delaying the progression of DKD.

### 2.4. Necroptosis

Necroptosis, a non-apoptotic form of PCD, exhibits characteristic morphological changes, including cytoplasmic granulation, organelle and/or cellular swelling, and the compromised integrity of the cytoplasmic membrane (Figure 5). Necroptosis is regulated by the receptor-interacting protein kinase 3 (RIPK3) and its effector, mixed lineage kinase domain-like (MLKL), which is initiated upon the interaction of RIPK3 with upstream adaptors such as receptor-interacting protein kinase 1(RIPK1), TIR domain-containing adaptor-inducing interferon-β (TRIF), or DNA-dependent activator of IFN-regulatory factors (DAIs). This process occurs in response to death receptor activation, toll-like receptor (TLR) signaling, pathogen invasion, or cellular stress in the absence of an external pathogen [139]. The modality of cell death is contingent upon the nature and magnitude of the inciting stimulus. Exposure to sublethal doses of various stressors such as heat, radiation, hypoxia, and cytotoxic anticancer drugs can elicit apoptosis in cells. However, at higher dosages, these same stimuli may precipitate necrosis [114].

Ubiquitin carboxyl-terminal esterase L1 (UCHL1) has been identified as playing a more prominent regulatory role in response to necroptotic processes, suggesting that necroptosis may significantly contribute to podocyte loss in DKD under the influence of UCHL1 regulation. Consequently, targeting UCHL1 to suppress the RIPK1/RIPK3 pathway represents a promising novel approach for preserving podocytes in DKD patients [140]. A recent investigation revealed that the total flavonoids of Abelmoschus manihot (TFA) exert a significant nephroprotective effect in DKD by suppressing necroptotic podocyte injury and RF. This is achieved through the inhibition of the TNF-α-induced signaling pathway involving RIPK1, RIPK3, and MLKL [141]. Despite relatively limited research on the impact of podocyte necrosis on DKD, there is an urgent need for comprehensive basic research to inform clinical therapeutic strategies.

### 2.5. Ferroptosis

Ferroptosis, originally defined in 2012 as an iron-dependent form of cancer cell death, distinct from apoptosis, necrosis, and autophagy [142], is characterized by the excessive accumulation of intracellular iron-ion-dependent lipid peroxides [143]. A primary mechanism of ferroptosis is associated with iron metabolism, lipid metabolism, and the accumulation of ROS [144]. In certain cell types, enzymes from the NOX family play a pivotal role in facilitating this process [145]. Key characteristics in the biological process of ferroptosis are principally identified through the assessment of iron death markers and the observation of lipid peroxidation events. During this process, mitochondrial morphology undergoes a distinct change, manifesting as a reduction in size and concomitant increase in density. Furthermore, the upregulation and translocation of transferrin receptor 1 (TfR1) to the cell membrane serve as key differentiating features for recognizing ferroptotic cells [146].

System Xc (-) is a cysteine/glutamate transporter responsible for facilitating the intracellular uptake of extracellular cysteine [147]. Following conversion by glutathione synthetase (GSS) and GCL (glutamate-cysteine ligase), cysteine is transformed into reduced glutathione (GSH) [148]. GPX4 utilizes GSH to mitigate lipid peroxidation and prevent ferroptosis. Impairment in cysteine absorption or reduction in GPX4 protein levels heightens the risk of ferroptosis [149]. The iron ions (Fe^3+^) enter the cells through binding with transferrin (TF), subsequently undergoing the Fenton reaction to convert to Fe^2+^, which generates hydroxyl radicals [150]. These radicals initiate lipid peroxidation and ROS production, ultimately driving ferroptosis (Figure 6).

Researchers have identified that HG contributes to podocyte ferroptosis [151], which in turn triggers the primary pathological changes leading to renal dysfunction deterioration and exacerbation of DKD progression [144]. Furthermore, it has been discovered that germacrone plays a role in podocyte ferroptosis, while the expression of mmu_mmu_circRNA_0000309, encoded within the host gene vascular endothelial zinc finger 1 (*VEZF1*), exhibits a significant decrease in DKD mice. Germacrone in DKD mice, by restoring the level of mmu_circRNA_0000309, a circRNA that sponges miR-188-3p, has shown potential in rescuing podocyte dysfunction and leads to increased glutathione peroxidase 4 (GPX4) expression. This ultimately counteracts the ferroptosis-dependent mitochondrial function and podocyte apoptosis. For this reason, targeting the regulation of the mmu_circRNA_0000309/miR-188-3p/GPX4 signaling pathway could further strengthen the therapeutic potential of germacrone for DKD treatment [152].

The recent groundbreaking study has revealed a significant increase in CD44 protein expression in podocytes when exposed to HG levels, which is linked to the development of renal fibrosis. CD44 influences epigenetic adaptability by regulating iron endocytosis and contributing to ferroptosis. Intriguingly, the identification of novel ferroptosis markers, CD44 and zinc finger protein 36 (ZFP36), suggests their potential role as mediators in the interaction between immune-mediated oxidative stress (IME) and ferroptosis in DKD [153].

Notably, the recent experiment demonstrated that specificity protein 1(Sp1) is capable of binding to the GC-box in the Prdx6 promoter, thereby directly regulating the transfection-induced activation of Prdx6 in podocytes under HG conditions. This mechanism potentially involves the suppression of oxidative stress and the prevention of ferroptosis [154]. Additionally, a separate study revealed that treatment with Tanshinone IIA (TIIA) effectively mitigated HG-induced podocyte injury and ferroptosis, partly through targeting the embryonic lethal abnormal visual-like protein 1/acyl-coenzyme A synthetase long-chain family member 4 (ELAVL1-ACSL4) axis, thus providing a promising therapeutic target for DKD treatment [151].

Furthermore, a recent study has highlighted the promising potential of utilizing bone marrow mesenchymal stem cell-derived exosomes (BMSC-Exo) as a vector for delivering miR-223-3p, in order to ameliorate ferroptosis triggered by the HBV-X protein in podocytes. The mitigating impact of miR-223-3p is achieved through its targeted suppression of histone deacetylase 2 (HDAC2), ultimately resulting in decreased STAT3 phosphorylation [155]. Therefore, BMSC-Exo show promise as a vector for delivering relevant molecules precisely to podocytes, thereby suppressing the high-glucose-induced ferroptosis process and protecting DKD from further deterioration. Future research should delve deeper into the role of ferroptosis in podocytes in DKD. Additionally, it has been verified that PKC expression increases in the podocytes of DKD [156].

A study has demonstrated that in Lund human mesencephalic (LUHMES) cells in Parkinson’s disease, the activation of PKC activates MEK to promote ferroptosis [157]. Furthermore, it has been confirmed that inhibiting PKCα leads to a reduction in body iron content in diabetes [158]. It is plausible to suggest that PKC activation contributes to an elevated iron burden in podocytes within DKD, concurrently triggering the MEK pathway in podocytes, ultimately facilitating iron-dependent ferroptosis. Therefore, iron chelators, Fer-1 derivatives, and PKC inhibitors may be potential drug candidates for DKD.

### 2.6. Others

Cuproptosis is a recently identified form of regulated cell death characterized by excessive Cu^2+^. Mitochondria are the primary targets, as copper-induced oxidative damage to the mitochondrial membrane and disruption of the tricarboxylic acid cycle (TCA) enzymes are observed. Copper binds to lipoylated components within the TCA cycle, leading to the aggregation of copper-bound lipoylated mitochondrial proteins and subsequent depletion of Fe-S clusters, which induces proteotoxic stress and ultimately results in cell demise [159]. Disturbance of renal copper homeostasis may drive the progression of DKD, and selective Cu(II) chelation can inhibit the activation of abnormal TGF-β1, thereby slowing down the process of RF and mitigating the progression of DKD. Empirical research indicates a significant rise in urinary copper levels in DKD with microalbuminuria, and the concentration of copper in the blood is believed to be associated with the progression of DKD [160]. A recent study utilizing bioinformatics techniques successfully identified and validated three hub genes associated with immune and cuproptosis-related genes for DKD. One such gene is follistatin-like protein 1 (*FSTL1*), mainly expressed in podocytes and glomerular mesangial cells [161]. Nonetheless, empirical evidence has yet to confirm this association. Given the current scarcity of research, we anticipate future studies will significantly contribute to this field.

Podoptosis, characterized by cytoplasmic vacuolization, endoplasmic reticulum stress, and dysregulated autophagy at the morphological level, is associated with cellular instability. Murine double minute (MDM2) plays an essential role in maintaining cellular homeostasis and ensuring the long-term viability of podocytes by inhibiting podoptosis, a p53-regulated cell death process with previously unclear features. In an animal model of DKD, decreased MDM2 expression alleviated the suppression of p53 before and during progressive glomerulosclerosis, ultimately leading to podoptosis-induced loss of podocytes and proteinuria [162]. Therefore, it is reasonable to infer that podoptosis serves as a potentially pivotal factor in the progression of DKD, particularly in relation to the loss of podocytes, generation of proteinuria, and development of glomerular fibrosis.

## 3. Potential Therapeutic Targets for Preventing Podocyte Death in DKD

### 3.1. Potential Preclinical Therapeutic Approaches Targeting Podocyte Death

As previously discussed, hyperglycemia induces pathological damage to podocytes through multiple pathways, including oxidative stress and epigenetic regulation. As universally acknowledged, one of the five pillars of diabetes management is exercise therapy. Regular physical activity has been shown to ameliorate DKD, potentially due to the protective effect of irisin, a hormone secreted by skeletal muscles, in mitigating diabetes-induced kidney damage. This protective mechanism involves the suppression of excessive activation of the PI3K/AKT/mTOR-signaling pathway, thereby facilitating the restoration of autophagy in podocytes [163]. Maintaining a consistent circadian rhythm is crucial for mitigating the progression of DKD. Research suggests that autophagy, a process regulated by the circadian clock, plays a pivotal role in preserving podocyte viability. Disruption of the circadian regulation of autophagy leads to podocyte damage and proteinuria [164].

The accumulation of cholesterol esters and fatty acid metabolites within podocytes in the DKD model is related to the pathogenesis of glomerular dysfunction [165], serving as pivotal mediators in renal lipid accumulation, inflammation, oxidative stress and fibrosis [166]. The quantity and quality of lipids are correlated with renal injury caused by lipotoxicity, which has been observed to induce podocyte injury [167]. In adiponectin-deficient mice, compromised podocyte-specific Adenosine 5′-monophosphate (AMP)-activated protein kinase (AMPK) leads to disrupted cellular morphology, foot process effacement, and subsequent development of proteinuria [168]. The *ACACB* gene, which encodes the rate-limiting enzyme for fatty acid oxidation, is expressed in podocytes and renal tubular epithelial cells in mice. Researchers have validated that in Chinese patients, the ACACB SNP rs2268388 and associated changes in fatty acid oxidation may significantly contribute to increased susceptibility to advanced T2DN, with a 2-fold or greater risk per risk-variant copy [169]. Importantly, the epigenetic effect of Acetyl-CoA synthetase 2(ACSS2) enhances raptor expression through H3K9 acetylation, thereby activating the mammalian mTORC1 pathway. Conversely, inhibiting ACSS2 leads to increased autophagy and reduced podocyte damage [170]. Therefore, targeting key lipid metabolic enzymes by modulating the AMPK-mTOR pathway might represent a novel therapeutic approach for modulating podocyte death mechanisms.

During the process of HG-induced podocyte injury, specific molecules and signaling pathways are implicated. Under HG conditions, EGFR, a receptor tyrosine kinase, becomes activated, leading to an upregulation of the autophagy inhibitor, rubicon. This is accompanied by a decrease in beclin-1 expression and the inhibition of LC3B autophagosome formation. These alterations directly contribute to the inhibition of autophagy, exacerbating podocyte damage and increasing proteinuria, ultimately accelerating the progression of DKD [171]. In DKD, a decline in vitamin D receptor (VDR) expression and autophagy were observed. Supplementation with Calcitriol demonstrated a protective impact on renal function in rats by significantly attenuating renal injury and alleviating HG-induced cellular stress in podocytes. This effect was achieved through the regulation of the VitD3/VDR signaling pathway and modulation of Atg16L1 expression as a downstream target [172]. The transient receptor potential canonical 6 (TRPC6) is a calcium-permeable ion channel protein involved in the regulation of intracellular calcium ions. A recent biopsy study on diabetic patients has demonstrated a significant correlation between elevated TRPC6 expression, reduced calpastatin levels, compromised autophagy, and podocyte injury. Experimental evidence from in vivo studies has revealed that diabetes-induced upregulation of TRPC6 in podocytes was accompanied by a decline in autophagic activity. Successful restoration of autophagy was achieved through transgenic overexpression of calpastatin and pharmacological intervention targeting calcium proteases, which led to a decrease in nephrin loss and prevention of albuminuria in diabetic mice [173]. Consequently, targeting TRPC6 and/or calcium proteases to restore renal glomerular autophagy shows promise as a potential therapeutic approach for DKD. Moreover, a more focused strategy towards podocytes would likely minimize treatment-related morbidity. When administered individually or in combination, empagliflozin and linagliptin exhibited a notable increase in the volume density of autophagosomes and autolysosomes within podocytes. Both drugs prevented foot process effacement and attenuated urinary albumin excretion, thereby contributing to a deeper understanding of the renal protective mechanisms associated with sodium glucose cotransporter 2 (SGLT2) and dipeptidyl peptidase-4 (DPP4) inhibitors in diabetes [174]. The synergistic senolytic approach utilizing dasatinib and quercetin effectively guards against the development of DKD by stimulating autophagy and preventing podocyte dedifferentiation through the Notch signaling pathway [175].

Autophagy and apoptosis, two distinct modes of cell death, can both contribute to podocyte injury through the modulation of the mTOR pathway. The selective inhibitor of the mTOR signaling, Rapamycin, effectively reduces abnormal podocyte demise and mitigates podocyte damage. Administration of Rapamycin results in a reversal of podocyte foot process effacement, accompanied by an increase in the number of autophagosomes and microtubule-associated protein light chain 3 (LC3)-positive podocytes. In diabetic mice, Rapamycin exhibits a nephroprotective impact by boosting autophagy and inhibiting podocyte apoptosis, demonstrating its therapeutic potential [176]. Spironolactone (SPL) employs a dual mechanism to prevent abnormal podocyte loss as well. On one hand, it upregulates angiotensin II (Ang II) levels, thereby enhancing autophagy. Specifically, SPL inhibits the PI3K/AKT/mTOR signaling pathway to promote autophagy [177]. On the other hand, SPL mitigates glucose-induced podocyte apoptosis by reducing ROS generation. In diabetes, elevated glucose levels stimulate the upregulation of serum and glucocorticoid-induced kinase-1 (Sgk1), subsequently activating NADPH oxidase through the mineralocorticoid receptor (MR) pathway, ultimately leading to increased ROS production and subsequent podocyte apoptosis, culminating in proteinuria [178]. Moreover, the expression of Bcl-2 was notably reduced in biopsy samples of renal tissue from patients with DKD. Wogonin exhibited a dose-dependent effect with a dual action of suppressing podocyte apoptosis and enhancing autophagy in a streptozotocin (STZ)-induced diabetic mouse model [179]. Uncommonly, tunneling nanotubes (TNTs), a cellular structure that necessitates TNF-alpha-induced protein 2 (TNFAIP2)/M-Sec, function as transient membrane conduits facilitating organelle transfer between cells. The upregulation of TNFAIP2 in human podocytes with DKD, induced by HG and AGEs, promotes the exchange of autophagosomes and lysosomes via TNTs, thereby alleviating AGE-induced autophagy, lysosomal dysfunction, and apoptosis. The protective role of TNFAIP2 in podocytes with DKD is underscored by its facilitation of TNT-mediated organelle transfer, suggesting it as a promising therapeutic target [180]. 

Recent studies have increasingly suggested that natural compounds have the potential to regulate ferroptosis in podocytes, showing promising prospects for the management of DKD [181,182]. As an illustration, Ginkgolide B (GB) has been found to exert a regulatory effect on the expression of ferroptosis-related marker GPX4, specifically suppressing its ubiquitination. Simultaneously, GB has demonstrated a suppressive impact on intracellular iron accumulation and ROS levels, effectively mitigating oxidative stress and preventing ferroptosis, ultimately contributing to the amelioration of DKD [183]. Furthermore, rhein exerted a suppressive effect on the expression of Rac Family Small GTPase 1 (Rac1) and its downstream targets, NADPH oxidase 1 (NOX1) and β-catenin, in podocytes exposed to HG conditions. Overexpression of Rac1 (oe-Rac1) resulted in a decrease in the synthesis of SOD, GSH, and GPX4, along with a reduction in the expression of Solute Carrier Family 7 Member 11 (SLC7A11) and nephrin, both crucial proteins in HG-treated podocytes. Consequently, rhein’s regulation of the Rac1/NOX1/β-catenin signaling axis contributes to the attenuation of DKD by inhibiting ferroptosis and EMT [184]. In addition, a recent investigation uncovered that the natural antioxidant pterostilbene exerts its protective effect by downregulating single-stranded DNA-binding protein (SSBP1) expression and subsequently inhibiting the DNA-PK/p53 signaling pathway, thereby ameliorating high fructose-induced glomerular podocyte ferroptosis and injury [185]. 

Lastly, recent studies have increasingly implicated oxidative stress as a central mediator in the response to diverse stimuli, with ROS and reactive nitrogen species (RNS) serving as key intracellular signaling molecules that facilitate autophagy [186]. Flavonoids have been shown to promote podocyte autophagy and restrain excessive RAAS activity by suppressing upstream oxidative stress and inflammatory pathways, ultimately ameliorating DKD. TGF-β1 signaling plays a regulatory role in this process [187]. Activation of the TGF-β1 pathway accelerates podocyte apoptosis, contributing to the progression of DKD [188]. Rhein-loaded polyethyleneglycol-co-polycaprolactone-co-polyethylenimine nanoparticles (PPP-RH-NPs) demonstrate a more pronounced effect in suppressing the abnormal expression of TGF-β1. It can be inferred that PPP-RH-NPs function by inhibiting the aberrant expression of TGF-β1, thereby mitigating podocyte apoptosis and improving DKD. Importantly, nanoparticles within the size range of 5–30 nanometers are capable of penetrating the glomerular filtration barrier and selectively target podocytes within the glomerulus [189], highlighting the potential for targeted renal delivery of nanocarriers for treating DKD.

### 3.2. Clinical Study

In a clinical trial, a total of 79 patients diagnosed with DKD were recruited and randomly allocated to either a control group or a group receiving Baoshenfang (BSF) treatment. Both groups received standard care for DKD, and after 12 weeks of intervention, the BSF treatment group demonstrated significantly reduced 24 h urinary protein excretion, as well as lower levels of serum creatinine and blood urea nitrogen compared to the control group. This study emphasizes the therapeutic potential of the BSF herbal formula in managing DKD, which is attributed to its mechanism of suppressing the NOX-4/ROS/p38 pathway, thereby alleviating oxidative stress and apoptosis in podocytes, while preventing proteinuria-induced cell damage [190].

A randomized, double-blind clinical study involving 42 diabetes patients was conducted, in which the participants were administered the maximum recommended dose of RAS inhibitors. The study divided the participants into two groups, with 21 individuals in each group. One group received GTP treatment for a duration of 12 weeks, while the other served as the placebo control. The findings showed a significant decrease of 41% in the urinary albumin-to-creatinine ratio (UACR) for the GTP group, whereas the placebo group experienced a marginal increase of 2% in UACR (*p* = 0.019). Consequently, it was inferred that GTP contributes to reducing podocyte apoptosis by activating the Wnt pathway and thereby leading to a reduction in proteinuria for diabetes patients using the maximum dose of RAAS inhibitors [191].

Limited clinical trials have provided evidence for the safety and feasibility of mesenchymal stem cell (MSC) therapies in managing DKD, suggesting a promising proof-of-concept [192]. To optimize the safety and efficacy of these therapeutic compounds, pharmaceutical studies are currently underway. Diabetic mice exhibited an increased p-mTOR/mTOR ratio, confirming their targeted delivery to affected organs. Mesenchymal stem cells (MSCs), representing a promising and emerging therapeutic approach, are currently under intense scrutiny and investigation. Elevated p62 levels, decreased ULK1 and autophagy-related 12 (Atg12), as well as a reduced LC3B/LC3A ratio in their kidneys, are consistent with diabetes-induced autophagy suppression. Following injection of human umbilical-cord-derived mesenchymal stem cells (hUC-MSCs), these changes were significantly reversed in the kidneys. The protective effect of hUC-MSCs on the kidneys was closely associated with the decline in circulating TGF-β1 levels and the restoration of autophagy function within the kidneys [193]. Placental MSCs (pMSCs) can effectively ameliorate renal damage and reduce podocyte injury in DKD rats by modulating the SIRT1/Forkhead box O1 (FOXO1) pathway, which in turn enhances podocyte autophagy [194].

## 4. Conclusions and Discussion

As the global prevalence of diabetes continues to rise, DKD has emerged as a major public health concern, reflecting the anticipated substantial financial burden on healthcare systems. The normal process of cellular death plays a crucial role in eliminating aged, damaged, or abnormal cells and maintaining the proper functioning of organs. However, disruption of this mechanism has the potential to initiate and advance various diseases. The primary mechanism driving the progression of DKD is injury and loss of podocytes. Podocyte death can occur through multiple mechanisms, including autophagy, pyroptosis, apoptosis, necroptosis, ferroptosis, and others. A comprehensive understanding of these mechanisms is essential for the development of pharmacological agents capable of effectively halting the progression of DKD.

While mechanisms of podocyte death have been identified in numerous experimental models of DKD, their direct clinical applicability and effectiveness in DKD remain incompletely understood and warrant further investigation. In addition, the clinical evidence for targeting the prevention of podocyte death in the treatment of DKD remains limited, hampering the application of various approaches in clinical trials. Consequently, there is an urgent need for high-quality and compelling clinical research to advance this field.

As noted above, while the aforementioned research mechanisms partially reveal the pathophysiological processes of podocyte death in the development of DKD, it is important to recognize that the underlying mechanisms are intricate and diverse. There exists a significant interplay among different death pathways in podocytes. In addition to this, oxidative stress and inflammatory responses are of paramount importance in multiple cell death. Hence, it is evident that various degrees of heterogeneous stimuli or pathological changes in the extracellular and intracellular environments may trigger distinct podocyte death, necessitating further clarification on their exact levels of activation.

Novel targeted treatment strategies have recently emerged for patients with DKD, specifically targeting the path of podocyte death. The techniques involved in MSC isolation and cultivation techniques hold great promise for future applications [195]. Moreover, advancements have been made in the development of nanoparticles targeting podocyte death as a therapeutic approach. When designing targeted medications, it is crucial to consider their pharmacokinetic properties. Similarly, the development of novel drug delivery systems requires specificity in targeting podocytes and the controlled release of the drug payload. However, the precise advantages of employing this approach to individualize treatment intensity are yet to be fully substantiated through rigorous validation.

Eventually, investigating the role of podocyte death in DKD may also have implications for other kidney diseases. Podocyte injury and loss are prevalent characteristics of various glomerular diseases, extending beyond just DKD. Elucidating the common mechanisms underlying podocyte death across diverse kidney diseases could pave the way for targeted therapeutic interventions applicable to a broader range of patients.

Collectively, future research efforts will concentrate on elucidating the specific pathophysiological mechanisms underlying podocyte death in DKD, investigating the interplay of diverse podocyte death pathways in DKD, and pinpointing the precise conditions that trigger distinct podocyte death, with the ultimate goal of developing targeted pharmacotherapies for podocytes.

## Figures and Tables

**Figure 1 ijms-25-09035-f001:**
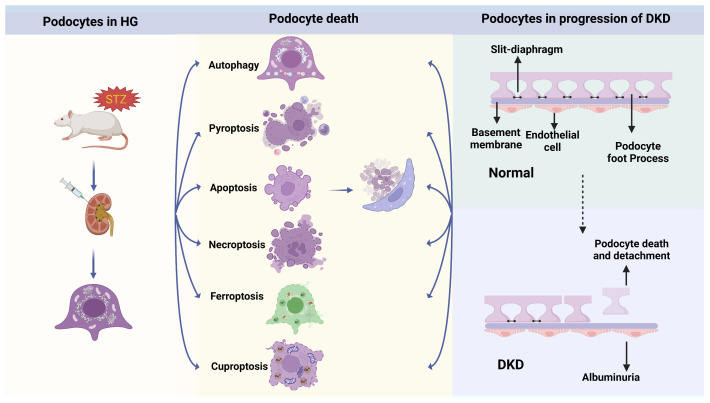
Podocytes undergo distinct cell death in a high-glucose (HG) environment, which disrupts their cellular structure. Within the kidney, the glomerular filtration barrier (GFB) is composed of three distinctive cellular layers, arranged in an inner-to-outward sequence: fenestrated endothelial cells, the GBM, and podocyte foot processes. The abundant fenestrated endothelial cells serve as the first barrier of GFB. The second barrier is formed by the GBM, which is jointly synthesized and secreted by endothelial cells and podocytes. The third barrier is established through the adherent junction of podocyte foot processes, called slit diaphragms (SDs), ensuring selective blood filtration. Damaged podocytes, showing foot process effacement, experience detachment from the basement membrane, resulting in developed proteinuria. Created with BioRender.com.

**Figure 2 ijms-25-09035-f002:**
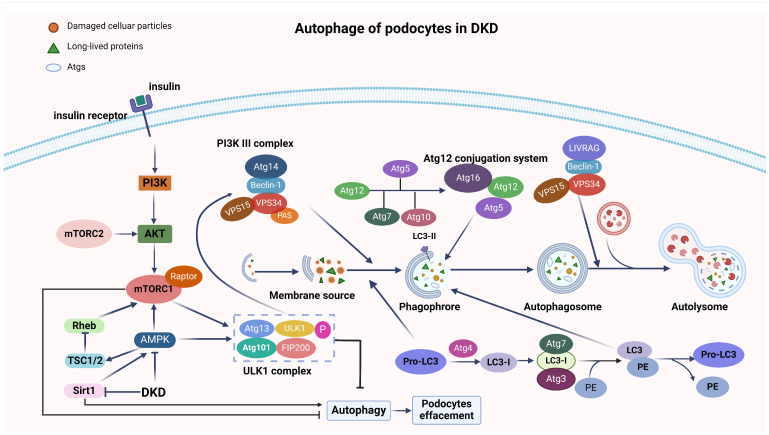
Under HG conditions, the activation of mTOR inhibits autophagy by suppressing the ULK1 complex. Conversely, AMPK negatively modulates mTOR activity, thereby promoting the induction of autophagy. The PI3K III complex is essential for the induction of autophagy. The *Atg* genes regulate autophagosome formation through the *Atg12*–*Atg5* and LC3-II complex. *Atg12*, involving *Atg7* and *Atg10*, binds to *Atg5* in a ubiquitin-like reaction, forming a large complex that promotes autophagosome synthesis. Pro-LC3 is converted to cytosolic LC3-I by the action of the Atg4 protease. LC3-I, with the participation of *Atg7* and *Atg3*, attaches to phosphatidylethanolamine (PE) to form lipidated LC3, or LC3-II, which adheres to the autophagosomal membrane, regulating various steps in autophagosome formation. mTOR: mechanistic target of rapamycin; PI3K: phosphoinositide 3-kinase; AMPK: AMP activated protein kinase; *Atg*: autophagy-related genes; RAS (renin–angiotensin system); Rheb (Ras homolog protein enriched in brain); TSC1 (Tuberous Sclerosis Complex 1) or TSC2; ULK1 (UNC-51-like kinase 1); VPS (vascular permeability factor); FIP200 (focal adhesion kinase family interacting protein of 200 kDa); PE(phosphatidylethanolamine); LC3 (microtubule-associated protein 1 light chain 3); PAS (pre-autophagosomal structure). Created with BioRender.com.

**Figure 3 ijms-25-09035-f003:**
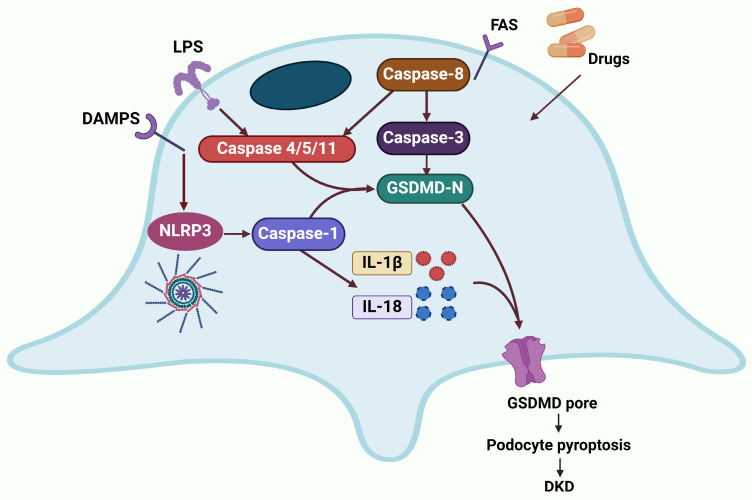
Pyroptosis primarily occurs through three main pathways: inflammasomes induce the cleavage of gasdermin D (GSDMD) or inflammatory precursors (such as pro-IL-1β and pro-IL-18), via caspase-1 activation, resulting in the formation of membrane pores and subsequent pyroptotic activation. Caspase-4/5/11 directly senses bacterial lipopolysaccharides (LPS) and cleaves GSDMD, leading to the generation of its N-terminal fragment, which promotes cell membrane pore formation, causing membrane rupture and subsequent activation of pyroptosis via non-inflammatory body pathway. TNF-α-activated caspase-8 facilitates the cleavage of GSDMD, generating GSDMD-N, which ultimately drives the process of pyroptosis. Created with BioRender.com.

**Figure 4 ijms-25-09035-f004:**
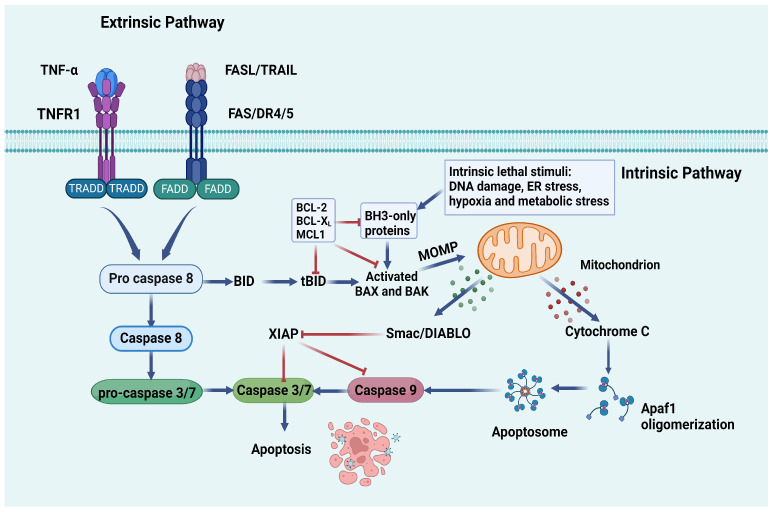
A schematic illustration of apoptosis mechanism. The activation of the intrinsic apoptotic pathway occurs when mitochondrial function is disrupted in response to various stress-inducing factors, including DNA damage, endoplasmic reticulum (ER) stress, and reactive oxygen species [118]. The key regulators of the intrinsic apoptosis pathway are the B-cell lymphoma-2 (Bcl-2) family proteins, which consist of pro-apoptotic Bcl-2 homology-3 (BH3)-only proteins (Bcl-2 interacting mediator of cell death (BIM), BH3-interacting domain death agonist (BID), p53-upregulated modulator of apoptosis (PUMA), Bcl-2-modifying factor (BMF), phorbol-12-myristate-13-acetate-induced protein 1 (PMAIP1, also known as NOXA), BCL-2-interacting killer (BIK), BCL-2-associated agonist of cell death (BAD)), activator of apoptosis harakiri (HRK), prosurvival proteins (BCL-2, BCL-2-like 1 (BCL-XL), BCL-2-like 2 (BCL-W), myeloid cell leukemia-1 (MCL-1), BCL-2-related protein A1 (A1/BFL-1), and apoptosis effectors (BCL-2-associated X protein (BAX), BCL-2 antagonist/killer 1 (BAK), and BCL-2-related ovarian killer (BOK)) [119,120]. Under normal physiological conditions, the pro-survival BCL-2 proteins can bind and inhibit the effectors of apoptosis to prevent PCD [121]. BH3-only proteins exhibit a high binding affinity to the prosurvival proteins [122]. Upon activation, BAX and BAK form oligomers, leading to a change in the mitochondrial membrane potential (MOMP), subsequently releasing cytochrome C and the second mitochondria-derived activator of caspases/direct inhibitor of apoptosis-binding proteins with low pI (Smac/DIABLO) [123,124]. The released factors facilitate apoptosis by activating the caspase cascade [125], ultimately resulting in the proteolytic cleavage of numerous proteins, culminating in cell death [126,127]. The activation of the extrinsic pathway is initiated by death receptors within the TNF receptor superfamily, such as focal adhesion complexes (FAS) receptor (CD95/APO-1), tumor necrosis factor receptor (TNFR1) and tumor necrosis factor (TNF)-related apoptosis-inducing ligand (TRAIL) receptors (DR4/DR5) [128]. Upon binding of ligands to these membrane receptors, downstream signaling cascades are initiated. The process begins with receptor clustering, followed by recruitment of adapter proteins such as Fas-associating protein with a novel death domain (FADD) and TNFR1-associated signal transducer (TRADD), which form the death-inducing signaling complex (DISC) [129,130]. The formation of DISC activates caspase-8 (a member of the cysteine protease family), subsequently triggering a proteolytic cascade that leads to the degradation of critical proteins and ultimately cell demise [99,131]. Created with BioRender.com.

**Figure 5 ijms-25-09035-f005:**
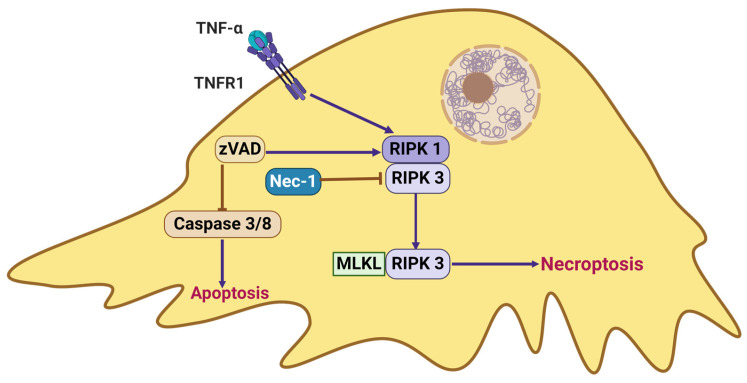
Necroptosis, a caspase-independent form of PCD, involves a process that requires mixed lineage kinase domain-like (MLKL) to undergo receptor-interacting protein kinase 3 (RIPK3)-dependent phosphorylation. This event facilitates the formation of pore complexes on the cellular membrane, ultimately resulting in cellular swelling and membrane rupture. Created with BioRender.com.

**Figure 6 ijms-25-09035-f006:**
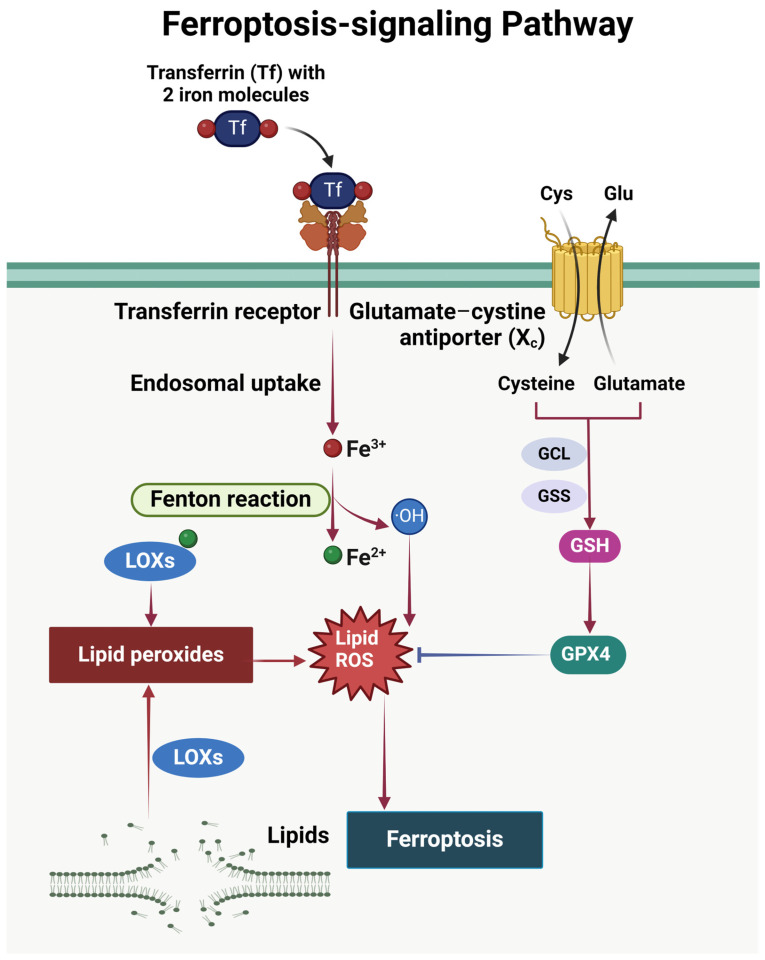
Excessive intracellular iron ions or impaired normal metabolism can lead to their accumulation. Within cells, iron ions may undergo oxidation by ROS to form trivalent iron (Fe^3+^), generating highly reactive free radicals, such as hydrogen peroxide (H_2_O_2_) and hydroxyl radicals (•OH). These free radicals can initiate oxidative reactions with biomolecules such as lipids, proteins, and nucleic acids, resulting in damage to cell membranes, organelles, and essential cellular molecules, This ultimately leads to cellular dysfunction and cell death. Created with BioRender.com.

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
