# Peer review of "Podocyte Death in Diabetic Kidney Disease: Potential Molecular Mechanisms and Therapeutic Targets"

_ijms, 2024, doi:10.3390/ijms25169035_

Round 1

Reviewer 1 Report

Comments and Suggestions for Authors

Summary statement

Zhong et al describe the known cellular and molecular mechanisms of podocyte cell death in DKD and discuss potential nonpharmacological and pharmacological therapies.

Strengths of the study

The authors performed an extensive review of the literature and provide a thorough analysis of the multiple causes of podocyte death and their roles in the pathogenesis of DKD. The figures are appropriate. The references are mostly current.

Areas to be improved

Major points

The translation to English is poor. There are many instances of sentence fragments, poor diction, run-on sentences, and grammatical errors. Also, some information is redundant in places, and the manuscript generally lacks flow and cohesiveness. The manuscript should be proofed and edited by a native English speaker.

Oftentimes, reasoning is unclear, context is either absent or incomplete, and connections are not fully made, which could be especially challenging for the nonexpert reader to understand. E.g., line 78: the roles of heparanse activity, glycocalyx loss, and progression to DKD is not clearly described.

Minor points

Line 625: UACR needs to be defined.

Line 629: RAS should be RAAS.

Comments on the Quality of English Language

Summary statement

Zhong et al describe the known cellular and molecular mechanisms of podocyte cell death in DKD and discuss potential nonpharmacological and pharmacological therapies.

Strengths of the study

The authors performed an extensive review of the literature and provide a thorough analysis of the multiple causes of podocyte death and their roles in the pathogenesis of DKD. The figures are appropriate. The references are mostly current.

Areas to be improved

Major points

The translation to English is poor. There are many instances of sentence fragments, poor diction, run-on sentences, and grammatical errors. Also, some information is redundant in places, and the manuscript generally lacks flow and cohesiveness. The manuscript should be proofed and edited by a native English speaker.

Oftentimes, reasoning is unclear, context is either absent or incomplete, and connections are not fully made, which could be especially challenging for the nonexpert reader to understand. E.g., line 78: the roles of heparanse activity, glycocalyx loss, and progression to DKD is not clearly described.

Minor points

Line 625: UACR needs to be defined.

Line 629: RAS should be RAAS.

Author Response

Comments 1: Oftentimes, reasoning is unclear, context is either absent or incomplete, and connections are not fully made, which could be especially challenging for the nonexpert reader to understand. E.g., line 78: the roles of heparanse activity, glycocalyx loss, and progression to DKD is not clearly described.

Response 1: Thank you for your kind suggestions. Below is an elucidation of the insufficiently delineated relationship between heparanse activity, glycocalyx loss, and progression to diabetic kidney disease (DKD).

The surface of endothelial cells is coated with a complex of glycoproteins referred to as the endothelial glycocalyx, primarily composed of heparan sulfate. In patients diagnosed with DKD, there is an elevation in renal heparinase activity, resulting in a reduction in heparan sulfate content and subsequent depth of the endothelial glycocalyx layer. Consequently, glomerular albumin permeability dependent on the endothelial glycocalyx increases among individuals with DKD, leading to augmented urinary protein excretion. This change can be found in line 84 to line 90.

Reviewer 2 Report

Comments and Suggestions for Authors

Manuscript “Cell Death of Podocytes in Diabetic Kidney Disease: Potential Molecular Mechanisms and Therapeutic Targets” presented by Zhong et al. summarizes current knowledge on molecular mechanisms of podocyte death during DKD as well as efforts in the development of specific pharmaceuticals for alleviation of the pathological processes. The work is information-rich and logically organized. The main topics are well-covered and referenced.

The major omission is the absence of specific statements on possible differences between described phenomena in relation to type 1 and type 2 diabetes.  Additionally, due to multiple biological models used in referenced research, it is often unclear how adequately described results represent DKD mechanisms developing in the human kidney.

Additional comments (Comments are made during continuous reading of the manuscript. Therefore, answers to some raised questions may occur later in the text.)

Title

1)      Podocytes are cells. Therefore, “Cell Death of Podocytes” sounds somewhat repetitive. “Podocyte death in “ is more concise.

Main text

1)     Some clarifications in the following can be made e.g. what are death patterns (podocytes or patients)? How can treatment be ideal?  “This review provides comprehensive insights into the various death patterns, underlying mechanisms,   and therapeutic strategies in  DKD. This will deepen our understanding of podocyte demise and lay the groundwork  for the development of novel, ideal treatment approaches for DKD.”

2)     Figure 2. Phagophore reportedly starts with a single-layer membrane. The figure depicts a situation where a double-layer membrane present at all stages. What is autophage? How top picture label “Autophage of podocytes in DKD” relates to figure legend “Figure 2. Under HG conditions, the activation of mTOR inhibits autophagy by suppressing  theULK1complex.”? Schematics have missing logical links between DKD and autophagy e.g. “The role of dysregulated autophagy in podocyte autophagy is a feature of DKD [76].”

 3)     Typo: Duringselective autophagy.

 4)  The following sounds as not supported by actual research: “The interplay between the UPS and the autophagy-lysosome pathway (ALP) in podocytes is crucial for the development of DKD, but current empirical evidence remains inconclusive,”.

Author Response

1. Point-by-point response to Comments and Suggestions for Authors

Comments 1: The major omission is the absence of specific statements on possible differences between described phenomena in relation to type 1 and type 2 diabetes.

Response 1: Thank you for your kind suggestions. The following explanation highlights potential differences in podocyte death linked to type 1 and type 2 diabetes.

The expression of genes associated with cell death, such as Bid, Dapk1, and Cd40, was significantly upregulated in diabetic podocytes. Furthermore, there was a slight increase in cleaved Caspase-3, indicating enhanced podocyte apoptosis in type 1 diabetic nephropathy (T1DM). Moreover, the decreased expression of genes primarily involved in RNA processing and endoplasmic reticulum function in diabetic podocytes suggests potential alterations in the mTOR pathway and autophagy/endoplasmic reticulum stress response, both of which are implicated in podocyte injury in type 2 diabetic nephropathy (T2DM). This change can be found in line 56 to line 63.

Comments 2: Additionally, due to multiple biological models used in referenced research, it is often unclear how adequately described results represent DKD mechanisms developing in the human kidney.

Response 2: Firstly, biological models such as mice and rats, which are easy to maintain and suitable for repeated experiments, exhibit a high degree of similarity to human DKD. Secondly, the biological processes of different species' diabetes models resemble the mechanisms observed in human DKD. For instance, when studying a specific cell signaling pathway, it is crucial to consider its role in DKD and its expression in human kidneys. Moreover, clinical drugs are typically initially tested in laboratory models to validate their effectiveness and safety before being investigated in humans. Finally, utilizing multiple biological models enables us to explore the consistency of results among these models and their complementary role in understanding the mechanisms of DKD, thereby contributing to a comprehensive understanding of the disease.

Comments 3: Podocytes are cells. Therefore, “Cell Death of Podocytes” sounds somewhat repetitive. “Podocyte death in “ is more concise.

Response 3: Agree. Thank you for your suggestion. We have revised the title from "Cell Death of Podocytes in Diabetic Kidney Disease: Potential Molecular Mechanisms and Therapeutic Targets" to "Podocyte Death in Diabetic Kidney Disease: Potential Molecular Mechanisms and Therapeutic Targets".

Comments 4: Some clarifications in the following can be made e.g. what are death patterns (podocytes or patients)? How can treatment be ideal?  

Response 4: Thank you for your question and suggestion. In this paper, the term "death patterns" refers to the various modes of podocyte death, such as autophagy, apoptosis, pyroptosis, necrosis, and ferroptosis. The pathophysiological changes in the development of DKD cannot be overlooked, including the reduction and disappearance of podocytes. Therefore, investigating the cell death mechanisms of podocytes and their underlying molecular regulatory mechanisms, targeting key regulatory factors to inhibit podocyte death, and preventing podocyte effacement are crucial in delaying the progression of DKD.

Comments 5: Figure 2. Phagophore reportedly starts with a single-layer membrane. The figure depicts a situation where a double-layer membrane present at all stages. What is autophage? How top picture label “Autophage of podocytes in DKD” relates to figure legend “Figure 2. Under HG conditions, the activation of mTOR inhibits autophagy by suppressing the ULK1 complex.”? Schematics have missing logical links between DKD and autophagy e.g. “The role of dysregulated autophagy in podocyte autophagy is a feature of DKD [76].”

Response 5: There is a "single-layer membrane" section added to the autophagy diagram in the text. The figure has been updated to explicitly show the relationship between podocyte autophagy, podocyte effacement, and the progression of DKD.

AMPK-mediated upregulation of autophagy has been discovered in mitochondria. In podocytes, au-tophagy is mainly regulated through the AMPK-ULK1 axis rather than inhibition of MTOR.

Autophagy is a highly conserved cellular mechanism by which cytoplasmic constituents including proteins and organelles are transported to lysosomes for degradation and preserving cellular homeostasis. It is a multistep process that involves the formation of isolation membrane, extension, formation of autophagosome, and final fusion with lysosomes to degrade phagocytic materials.

Comments 6: Typo: Duringselective autophagy.

Response 6: Thanks for bringing this to my attention. It has been modified. This change can be found in line 226.

Comments 7: The following sounds as not supported by actual research: “The interplay between the UPS and the autophagy-lysosome pathway (ALP) in podocytes is crucial for the development of DKD, but current empirical evidence remains inconclusive,”.

Response 7: UPS and autophagy are two major ways of protein degradation in cells. the autophagy lysosome pathway is an important compensation mechanism for mediating the degradation of ubiquitinated protein aggregates. In addition, proteasome inhibitors can also directly act on autophagy-related proteins to promote autophagy. Inhibition of proteasome leads to a compensatory stimulation of autophagy, whereas the inhibition of autophagy activates or impairs proteasomal flux via several mechanisms. Additionally, mitochondria have been described to be involved in the sensing of UPS impairment and in AMPK-mediated upregulation of autophagy. In podocytes, au-tophagy is mainly regulated through the AMPK-ULK1 axis rather than inhibition of MTOR [103]. Previous research has demonstrated that the deubiquitinating enzyme USP11 promotes autophagy by activating the AMPK/Akt/mTOR signaling pathway, leading to ULK1 activation and initiation of autophagy, thereby playing a role in reg-ulating autophagy in colorectal cancer.

2. Response to Comments on the Quality of English Language

Point 1: The translation to English is poor. There are many instances of sentence fragments, poor diction, run-on sentences, and grammatical errors. Also, some information is redundant in places, and the manuscript generally lacks flow and cohesiveness. The manuscript should be proofed and edited by a native English speaker.

Response 1: Thank you for your constructive criticism. I have carefully reviewed and revised the entire article.

Reviewer 3 Report

Comments and Suggestions for Authors

This review is focused specifically on podocyte cell death in diabetic kidney disease. Despite this focus the article will be useful reading for anyone studying disease, disease treatments and tissue engineering as cell death occurs under a large range of circumstances.

Importantly this article highlights the many different types of cell death and is a great read for anybody interested in expanding their knowledge in the cell death field. The review cover six different types of cell death and although the authors provide details of cell death in the context of podocytes the references provide information on studies that will be useful to a broader, non-renal, audience.

There are many manuscripts detailing different types of cell death and some very good reviews, but this review adds to the literature by placing cell death into the context of disease at different levels of detail. For example, high glucose->diabetes -> renal disease due to diabetes. Since the high glucose effect on renal endothelial cells applies to endothelial cells of other organs beside the kidney, this review is accessible to scientists and students from multiple disciplines. 

This review also covers potential therapeutic targets via the cell death pathways described. This section is brief but provides a paradigm for future reading and development of therapies.

There is good use of figures summarizing the pathway components for each type of cell death described making it easier to follow the text. The references are appropriate and contain many recent papers.

The manuscript is well written but some typos/tense mistakes are present. Read through the manuscript carefully to correct these. As an example;

line 71: change 'a prominent' to 'is a prominent'

line 313: change 'release' to 'released'

Author Response

1. Response to Comments on the Quality of English Language

Point 1: The manuscript is well written but some typos/tense mistakes are present. Read through the manuscript carefully to correct these. As an example;

line 71: change 'a prominent' to 'is a prominent'

line 313: change 'release' to 'released'

Response 1: Thank you for pointing this out. We agree with this comment. The language of the article has also been meticulously revised. We have revised 'a prominent' to 'is a prominent', which can be found in line 78.  We have revised 'release' to 'released', which can be found in line 328.
